# Volumetric study of the maxillary sinus in patients with sinus pathology

**Mario Pérez Sayáns** [1,2]*, **Juan A. Suárez Quintanilla**[3], **Cintia M. Chamorro Petronacci**[2], **José M. Suárez Peñaranda**[4], **Pía López Jornet**[5], **Francisco Gómez García**[5], **Yolanda Guerrero Sánchez**[6]

**1** Oral Medicine, Oral Surgery and Implantology Unit (MedOralRes Group), Faculty of Medicine and Dentistry, Universidade de Santiago de Compostela, Santiago de Compostela, Spain, **2** Health Research Institute Foundation of Santiago (FIDIS), Santiago de Compostela, Spain, **3** Area of Human Anatomy and Embryology, Faculty of Medicine and Dentistry, University of Santiago de Compostela, Santiago de Compostela, Spain, **4** Pathological Anatomy Service, University Hospital Complex of Santiago (CHUS), Santiago de Compostela, Spain, **5** School of Dentistry, Faculty of Medicine, Research Virgen de la Arrixaca Clinical University Hospital, IMIB-Arrixaca, Clínica Odontológica Universitaria Hospital Morales Meseguer, University of Murcia, Murcia, Spain, **6** Area oh Human Anatomy and Psychobiology, University of Murcia, Murcia, Spain

* perezsayans@gmail.com

**Data Availability Statement:** All relevant data are within the manuscript.

**Funding:** The author(s) received no specific funding for this work.

## Abstract

### Objectives

The aim of this study is 1) to obtain the area and volumes of the maxillary sinuses in patients affected by clinically unilateral sinus pathology by comparing the results to the contralateral sinus and 2) to determine the importance of the volumetric measures when diagnosing the percentage of sinus obliteration.

### Materials and methods

A single-centre observational retrospective clinical study was conducted in 214 patients with clinically unilateral sinus pathologies. Linear (mm), area (mm2) and volume (mm3) measurements were taken from Cone Beam Computed Tomography (CBCT) images of the affected sinus as well as from the contralateral ones. Histopathological study was performed using haematoxylin/eosin and PAS or Groccot stains. The lesions were classified into non-specific sinusitis, polyps, inverted papilloma, fungal sinusitis, cysts, mucocele and other lesions. Chi-squared test, ANOVA for independent samples and Pearson test were used for the statistical analysis.

### Results

A total of 100 sinuses were measured in 50 patients (28 men and 22 women, with an age of 43.6 years (SD = 18.3), 50 pathological and 50 healthy contralateral sinuses. The three-dimensional occupation volume of the affected sinuses was 97.1 mm$^3$ (62.5%) vs. 40.6 mm$^3$ (22.8%) in the healthy ones (p<0.0001). The medial-lateral width of the sinus in the frontal plane was significantly higher in the cysts group (32.4 mm, CI: 23–41.8 mm).

**Competing interests:** The authors have declared
that no competing interests exist.

## Conclusion

In medical terms, the global percentage of occupation determined using the classic manual
determination method does not differ from the three-dimensional percentage calculated
using specific complex software.

## Introduction

The quantification of the measures, areas and volumes of the maxillary sinus can be a complex
and tedious process, nonetheless, it is very useful for specifically determining its affectation
and/or level of obliteration [1]. The sinus morphology, in a geometrical sense, can look like a
sphere or a pyramid, therefore elaborating mathematical algorithms for its automatic calcula-
tion is a complex process. Anagnostopoulou et al. [2] arranged the sinuses into 4 classes,
according to their similarity to: semi-ellipsoid (type a: 15%); paraboloid (type b: 30%); hyper-
boloid (type c: 47%) and conical (type d: 8%). The anatomic dimensions of different structures,
including paranasal sinuses, can be measured using computed tomography images (CT) [3].

The anatomic features observed in reconstructed CT images confirm the variability in
terms of the shape of the maxillary sinus. The differences between the sinus areas and volumes
have proved to be of interest, not only to otorhinolaryngologists or radiologists, in terms of the
involvement of the ostium area which enables drainage into the nasal cavity, but also to odon-
tologists and maxillofacial surgeons who are intending to perform bone graft and sinus eleva-
tion procedures and who look to evaluate the sinus involvement in processes of odontogenic
origin [4].

The traditional techniques used for studying the paranasal sinus pathology include the
Sino-nasal outcome test (SNOT-22) [5] and Lund-Mackay [6] and Zinreich [7]'s modified
staging systems. Due to its hidden location within the facial skeleton, it is difficult to measure
the volume of the paranasal sinus in general clinical practice, meaning therefore that it is diffi-
cult to determine the severity of the pathology. The procedure is complex and requires experi-
ence in the use of radiological methods and computer programs. An imprecise sinus contour
measurement can have an impact on the estimation of the volume and the automatically calcu-
lated areas. Some authors have praised the use of linear measurements as a simplified method
for determining volumes [8] [1]

The evolution, growth and pneumatization of the maxillary sinus is complex and it varies
depending on the patient's age and gender. The reduction in the global volume may be the
first sign of aplasia or hypoplasia in craniofacial malformations of the first and second arch.
Several sinus pathologies exist in adults with the most prevalent being unspecific infections
and/or inflammations, although they can also be affected by benign and malignant tumour
pathologies. During the process of occupation of the maxillary sinus, the sinonasal organ
allows for the bilateral involvement of the sinuses even when only unilateral symptomology is
clinically present [9].

Our hypothesis is that sinus areas and volumes are not affected by the presence of pathol-
ogy, however they are affected by the obliteration percentage. The aim of this study is 1)
to obtain the area and volumes of the maxillary sinuses in patients affected by clinically uni-
lateral sinus pathology by comparing the results to the contralateral sinus and 2) to determine
the importance of the volumetric measures when diagnosing the percentage of sinus
obliteration.

## Material and methods

### Selection of patients

A single-centre observational retrospective clinical study was conducted with patients who attended the University Hospital Complex of Santiago de Compostela (CHUS) suffering from sinus pathologies during the period of 2009–2019. The multi-sliced computed tomography images (MSCT (Philips, Madrid, Spain)) of the patients who met the inclusion criteria were retrospectively studied.

**Inclusion and exclusion criteria.**   The inclusion criteria were as follows: male and female patients over the age of 18 years who attended a medical consultation due to a unilateral para-nasal sinus pathology which required a 3D radiological study and for at least one confirmatory biopsy or surgical treatment of the lesion to be performed. The exclusion criteria were: patients who had not signed the informed consent form, developing patients, patients with lesions localized in a region other than the maxillary sinus (i.e. the ethmoid or sphenoid sinuses, or the nasal cavity), lesions with bilateral clinical involvement, relapses of previously diagnosed and/or treated lesions, lesions which affect the maxillary sinus due to locoregional spread but which originate in a different location, lesions with odontogenic origin and malignant neoplasms.

A total of 214 patients with sinus lesions who had undergone a complete tomographic study were obtained, and of these 214 patients, 50 were selected as a representative sample by means of a systematic sampling using the Epidat 4.2 software (Epidat, SERGAS—Galician Public Health System, Santiago de Compostela, Spain), with a 23.4% probability of selection. The unit of analysis was the sinus, and the patients were therefore divided into two groups: sick/affected sinus and healthy/non-affected sinus.

All of the procedures carried out during this study complied with the ethical standards of the institutional and research committee and with the declaration of Helsinki of 1964 and its subsequent amendments. All of the patients signed the informed consent form in order to participate in the study, and likewise, they gave permission for the research results to be published anonymously. This study was approved by the Ethical Committee of the Autonomous Region of Galicia (Ref. No. 2019/596) and was carried out in accordance with the recommendations of the STROBE guide for observational studies.

### Clinical variables of the study

Participants' medical records were reviewed from July/2019 to October/2020. Data on the following variables was gathered from the study: date of birth, date of diagnosis, sex, whether the sinus was affected or healthy, the affected side of the sinus, and the histologically confirmed diagnosis. The sinus that was the reason for the medical consultation was considered as the pathological sinus and the contralateral sinus was considered as the control.

### Histopathological study

Through the histopathological diagnosis it was possible to determine more specifically the origin of the inflammation/infection, the type of tumour and/or the presence of other pathologies which had not been clinically diagnosed. This process was performed in a routine manner by means of paraffin inclusion and haematoxylin and eosin staining. Where necessary, multiple sections were taken from each of the blocks and PAS or Groccot stains were performed in those patients with a clinical suspicion of mycosis.

The final diagnosis was made by the clinical physician based on the results of the anamnesis and clinical examination, the specific radiological findings and the anatomopathological report.

The lesions were classified into: non-specific sinusitis, polyps, inverted papilloma, fungal sinusitis, cysts, mucocele and other lesions (fibrolipoma, cavernous haemangioma, mucosal thickening, granulomatosis with polyangiitis, organised clot).

## Radiological study

Cone Beam Computed Tomography (CBCT) was the radiological technique used. Linear (mm), area (mm$^2$) and volume (mm$^3$) measurements were taken from CT images of the affected sinus as well as from images of the contralateral sinus using the SECTRA v.3.3 medical software (SECTRA group, Teknikringen, Sweden). Due to the deep anatomical and structural analysis of the radiological studies the values of FOV were high, from 170–229 mm. Regarding the voxel size used, it was between 0.3–0.4 as a general range, adapted to each specific case, so there was absolute precision when calculating the volume. The following radiological variables were collected from the study: maximum anteroposterior length, medial-lateral length (maximal latero-lateral), maximum area of the sinus's contour, unoccupied sinus area, occupied sinus area, percentage of sinus occupation, sinus volume, unoccupied volume, and occupied volume. The linear and area measurements were taken in two planes: the horizontal plane and the frontal plane. The volume measurements were taken in the three planes of the space.

The anterior-posterior length and the medial-lateral length were measured in the horizontal plane at 5, 10, 15 and 20 mm above the palatine plate of the superior maxillary bone (Fig 1A). The highest value of the four measurements taken was considered as the maximum. The maximum areas of the sinus contour were measured, and likewise, the measurements of the non-

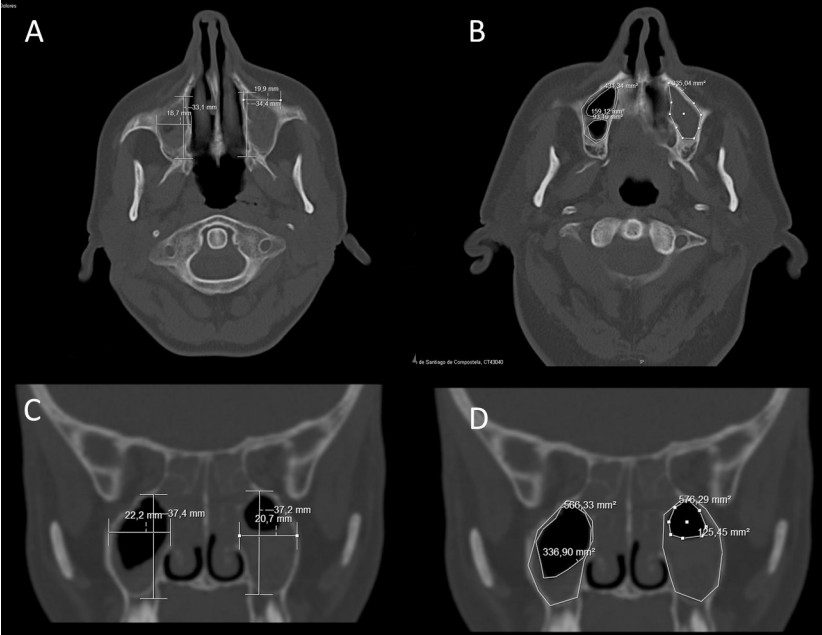

**Fig 1. Representation of the measurements taken.** A) Antero-posterior and medial-lateral length in the horizontal plane. Measurements taken at 5-10-15-20 mm from the palate plate or vault towards the cranial. B) Area of sinus contour, unoccupied area and affected area in the horizontal plane, at 5-10-15-20 mm cranially. C) Sinus height and width measured in the frontal plane. D) Area of sinus contour, unoccupied area and affected area in the frontal plane.

pathological/occupied areas were taken in the same 4 locations, considering the highest one (Fig 1B). The occupied area of the sinus was obtained using the difference between the maximum area of the sinus and the maximum unoccupied area.

In the frontal plane and for each of the cuts, the maximum height of the sinus and the medial-lateral width were measured (Fig 1C). In the cases in which said cut did not represent the maximum width, the maximum width from the corresponding cut and the correlative height for that same one were registered. In those same tomographic sections with maximum height and width in the frontal plane, the area of the sinus and the unoccupied area were registered in order to subsequently calculate the occupied area (Fig 1D).

The anteroposterior and medial-lateral measurements were taken perpendicularly to the plane, always considering the greatest distance of the anatomic limits. If intra-sinus septa were present, the areas were arithmetically added. The size of the unoccupied area of the sinus was determined taking into account the inner side of the Schneider's membrane.

The process of calculating the maxillary sinus volume was carried out using the Amira imaging analysis software (Thermo Fisher Scientific, London, United Kingdom). Once the Dicom files of the images which were to be studied had been obtained, the rendering volume of those files, as well as the three-dimensional image were obtained. Said image was subsequently used for the segmentation process of the maxillary sinus using AMIRA 5.6 (FEI, Visualization Sciences Group) in order to obtain more data. The LABELFIELD's module allowed for a surface model to be created that helped to both select neighbouring voxels (with Hounsfield' values over 700 in the case of bones and over 300 for blood vessels) and to differentiate soft tissue in a simple way. Likewise, the segmentation was firstly done manually and then semi-automatically by cutting the maxillary sinus into multiple 2D sections and assigning these regions to a volumetric structure. In the cases in which the maxillary sinus limit was unclear, the same criteria as those used in the linear area calculations were applied.

Different colours were used to determine the four studied structures: right maxillary sinus, right pathological process, left maxillary sinus, left pathological process. Once the image had been segmented, the volume calculation in mm$^3$ was automatically obtained, having marked at the beginning of each process, the size of the pixel or voxel used.

The radiological analysis was carried out independently by two operators (CMCP and MPS) These operators had previously been calibrated over 10 CT images different from the ones in the study and the Kappa index had been calculated, obtaining a degree of concordance between the measurements of 0.93. The final measurements were the result of the average of both observers.

## Statistical analysis

The data was analysed using the statistical program SPSS 23.0. The descriptive statistic was performed using frequencies and percentages for the categorical variables, and averages and standard deviations for the quantitative variables. The Kolmogorov-Smirnov test was used to verify the normality of the variables. Contingency tables were drawn up using the Chi-squared test to compare categorical variables. The analytical statistic was performed by comparing the variables using the ANOVA and t-student test for independent samples to compare means. The level of correlation between the quantitative variables was studied using the Pearson test. All of the divergences in which the p value was less than or equal to 0.05 were considered to be statistically significant (0.01 for the Pearson correlation coefficient).

## Results

A total of 100 sinuses were measured in 50 patients (28 men and 22 women, with an age range from 18 to 83 years and an average age of 43.6 (SD = 18.3), 50 pathological and 50 healthy

contralateral sinuses of the same patients. The descriptive data of the different variables can be found in Table 1. The most frequent pathology was non-specific sinusitis (48%), followed by polyps (24%), and fungal sinusitis (12%).

Globally, the maxillary sinuses, regardless of the existence of pathology, had the following average dimensions: anteroposterior length (horizontal plane) of 35.5 mm (SD = 3.9), medial-lateral length (horizontal plane) of 24.3 mm (SD = 3.8) and 25.6 mm (frontal plane) (SD = 4.3) and a height (frontal plane) of 38.2 mm (SD = 4.5). The maximum average area of the sinuses measured in the horizontal plane was 535.4 mm2 (SD = 123.9) and 624.5 mm2 (SD = 134.4) in the frontal plane. The global average of the three-dimensional volume was 147.6 mm3 (SD = 52). 70% of the studied sinuses presented a certain obliteration level, ranging from 6.9 to 100%. In 17 of the cases (17%) the obliteration was complete. The global average of the occupied area was 43.5% (SD = 38.6) in the frontal plane, 47.2% (SD = 41.4) in the horizontal

**Table 1. Descriptive data.** Descriptive data of all of the study variables classified according to sinus involvement.

| Variable | | N | % | | |
|---|---|---|---|---|---|
| Gender | Men | 56 | 56.0 | | |
| | Women | 44 | 44.0 | | |
| Diagnosis | Non-specific sinusitis | 48 | 48.0 | | |
| | Fungal sinusitis | 12 | 12.0 | | |
| | Inverted papilloma | 4 | 4.0 | | |
| | Polyp | 24 | 24.0 | | |
| | Cyst | 6 | 6.0 | | |
| | Other lesions | 6 | 6.0 | | |
| Affected side | Right | 50 | 50.0 | | |
| | Left | 50 | 50.0 | | |
| | | Min | Max | Av. | SD |
| Non-affected sinus | Age (years) | 18.2 | 83.0 | 43.6 | 18.3 |
| | Anteroposterior length in the horizontal plane (mm) | 50 | 26.7 | 44.8 | 35.7 |
| | Latero-lateral length in the horizontal plane (mm) | 16.4 | 32.8 | 24.5 | 3.4 |
| | Maximum sinus area in the horizontal plane (mm$^2$) | 339.3 | 908.8 | 542.2 | 114.3 |
| | Maximum unoccupied sinus area in the horizontal plane (mm$^2$) | 4.5 | 676.3 | 442.4 | 151.4 |
| | Maximum occupied sinus area in the horizontal plane (mm$^2$) | 4.8 | 492.3 | 99.8 | 137.7 |
| | Maximum height in the frontal plane (mm) | 28.2 | 50.9 | 38.0 | 4.5 |
| | Maximum width in the Frontal plane (mm) | 14.8 | 39.9 | 25.8 | 4.5 |
| | Maximum sinus area in the frontal plane (mm$^2$) | 283.2 | 1125.5 | 625.1 | 138.2 |
| | Maximum unoccupied sinus area in the frontal plane (mm$^2$) | 0.0 | 832.9 | 482.9 | 214.2 |
| | Maximum occupied sinus area in the frontal plane (mm$^2$) | 0.0 | 659.3 | 144.9 | 193.3 |
| | Total sinus volume (mm$^3$) | 52.3 | 254.0 | 141.1 | 42.25 |
| | Unoccupied sinus volume (mm$^3$) | 0.0 | 201.3 | 100.5 | 58.48 |
| | Occupied sinus volume (mm$^3$) | 0.0 | 150.0 | 40.6 | 50.44 |
| | Percentage occupation area in the horizontal plane (%) | 0.8 | 98.9 | 18.3 | 25.0 |
| | Percentage occupation area in the frontal plane (%) | 0.0 | 100.0 | 22.8 | 30.4 |
| | Percentage occupation volume (%) | 0.0 | 100.0 | 28.4 | 35.9 |

*(Continued)*

**Table 1.** (Continued)

| Variable | | N | % | | |
|---|---|---|---|---|---|
| Affected sinus | Age (years) | 18.2 | 83.0 | 43.6 | 18.3 |
| | Anteroposterior length in the horizontal plane (mm) | 23.2 | 45.9 | 35.3 | 4.3 |
| | Latero-lateral length in the horizontal plane (mm) | 12.3 | 31.8 | 24.0 | 4.2 |
| | Maximum sinus area in the horizontal plane (mm$^2$) | 181.2 | 896.4 | 528.7 | 133.7 |
| | Maximum unoccupied area in the horizontal plane (mm$^2$) | 0.0 | 499.8 | 119.6 | 167.9 |
| | Maximum occupied sinus area in the horizontal plane (mm$^2$) | 6.3 | 807.3 | 409.1 | 207.4 |
| | Maximum height in the frontal plane (mm) | 29.2 | 47.4 | 38.3 | 4.5 |
| | Maximum width in the frontal plane (mm) | 17.1 | 36.6 | 25.4 | 4.1 |
| | Maximum sinus area in the frontal plane (mm$^2$) | 372.7 | 917.3 | 623.9 | 131.9 |
| | Maximum unoccupied sinus area in the frontal plane (mm$^2$) | 0.0 | 756.0 | 215.9 | 244.0 |
| | Maximum occupied sinus area in the frontal plane (mm$^2$) | 4.5 | 871.5 | 408.0 | 273.7 |
| | Total sinus volume (mm$^3$) | 62.9 | 393.8 | 154.1 | 59.94 |
| | Unoccupied sinus volume (mm$^3$) | 0.0 | 245.7 | 57.1 | 57.92 |
| | Occupation sinus volume (mm$^3$) | 0.0 | 209.6 | 97.1 | 62.62 |
| | Percentage occupation area in the horizontal plane (%) | 1.2 | 100.0 | 76.1 | 33.7 |
| | Percentage occupation area in the frontal plane (%) | 0.7 | 100.0 | 64.2 | 39.3 |
| | Percentage occupation volume (%) | 0.0 | 100.0 | 62.5 | 35.2 |
| All sinuses | Age (years) | 18.2 | 83.0 | 43.6 | 18.2 |
| | Anteroposterior length in the horizontal plane (mm) | 23.2 | 45.9 | 35.5 | 3.9 |
| | Latero-lateral length in the horizontal plane (mm) | 12.3 | 32.8 | 24.3 | 3.8 |
| | Maximum sinus area in the horizontal plane (mm$^2$) | 181.2 | 908.8 | 535.4 | 123.9 |
| | Maximum unoccupied area in the horizontal plane (mm$^2$) ** | 0.0 | 676.3 | 281.0 | 227.1 |
| | Max. occupied sinus area in the horizontal plane (mm$^2$) ** | 4.8 | 807.3 | 254.4 | 234.2 |
| | Maximum height in the frontal plane (mm) | 28.2 | 50.9 | 38.2 | 4.5 |
| | Maximum width in the frontal plane (mm) | 14.8 | 39.9 | 25.6 | 4.3 |
| | Maximum sinus area in the frontal plane (mm$^2$) | 283.2 | 1125.5 | 624.5 | 134.4 |
| | Maximum unoccupied sinus area in the frontal plane (mm$^2$) ** | 0.0 | 832.9 | 349.4 | 264.9 |
| | Maximum occupied sinus area in the frontal plane (mm$^2$) ** | 0.0 | 871.5 | 276.5 | 270.2 |
| | Total sinus volume (mm$^3$) | 52.3 | 393.8 | 147.6 | 52.0 |
| | Unoccupied sinus volume (mm$^3$) ** | 0.0 | 245.7 | 78.8 | 61.9 |
| | Occupation sinus volume (mm$^3$) ** | 0.0 | 209.6 | 68.8 | 63.3 |
| | Percentage occupation area in the horizontal plane (%) ** | 0.8 | 100.0 | 47.2 | 41.4 |
| | Percentage occupation area in the frontal plane (%) ** | 0.0 | 100.0 | 43.5 | 40.7 |
| | Percentage occupation volume (%) ** | 0.0 | 100.0 | 45.4 | 39.3 |

SD = Standard deviation.

** Differences between affected and non-affected sinuses of p<0.001 level. Min: minimum; Max: maximum; Av: average.

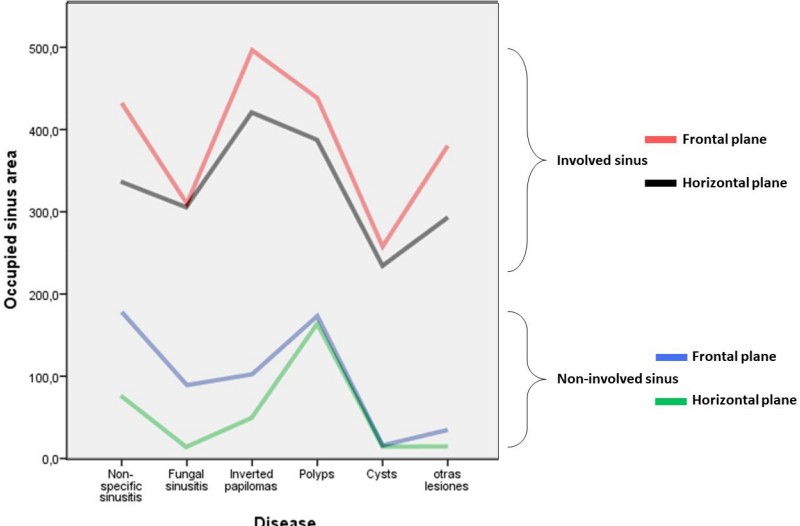

**Fig 2. Area occupation.** Diagram representing the average area of occupation in affected and healthy sinuses, measured in the horizontal or frontal plane, depending on the specific pathologies. In all of the pathologies, the occupation area was greater in the group of affected sinuses (p<0.001).

plane, and an average occupation volume of 45.4% (SD = 39.3) was observed in the three-dimensional measurements. No differences were observed between men and women, nor were they observed in the sinus volume or the occupied volume.

No statistically significant differences were found between the size of the affected sinuses and that of the healthy ones (in linear measurements, areas or volumes). However, differences were observed in the areas, volumes and percentages of occupation in all the planes. The affected sinuses presented occupation areas in the horizontal plane of 409.1 mm$^2$ vs. 99.8 mm$^2$ in the healthy ones (76.1% vs. 18.3%) and in the frontal plane of 408 mm$^2$ vs. 144.9 mm$^2$ (64.2% vs. 22.8%) (p<0.0001) (Fig 2). The three-dimensional occupation volume of the affected sinuses was 97.1 mm$^3$ (62.5%) vs. 40.6 mm$^3$ (22.8%) in the healthy ones (p<0.0001) (Fig 3).

The average percentage of occupation of the pathological sinus in the horizontal plane (76.1%), was positively correlated with the frontal plane value (64.2%) (CC = 0.712, p<0.0001)

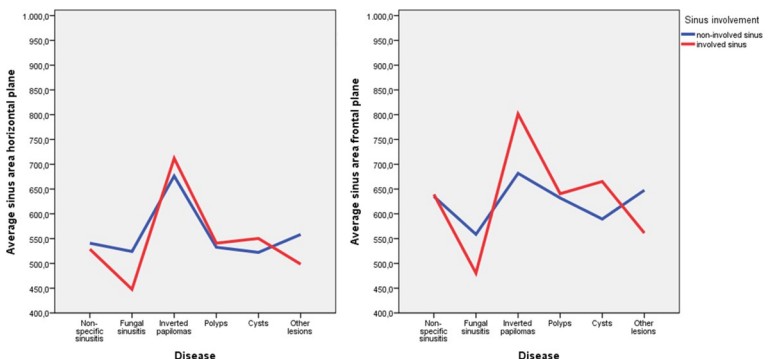

**Fig 3. Sinus contour.** Diagram representing the average sinus contour area in healthy and affected sinuses, depending on the specific pathologies. The inverted papilloma and the cysts were the lesions with a larger sinus area, p = 0.014 and p = 0.023 respectively.

**Table 2. Statistically significant differences in measurements and areas of the affected sinuses regarding the diagnosis.** The frontal width of the affected sinus is larger in the group of patients suffering from cysts and the sinus area measured in the frontal plane is larger in the group of patients with inverted papilloma.

| Sinus involvement | | | N | Average | SD | 95% Confidence interval | | P value |
|---|---|---|---|---|---|---|---|---|
| | | | | | | Lower | Upper | |
| Affected sinus | Maximum width in the frontal plane | Non-specific sinusitis | 24 | 25.3 | 3.7 | 23.8 | 26.8 | **Bonferroni test Cysts-NSS: p = 0.049 Cysts-FS: p = 0.026 Cysts-OL: p = 0.019** |
| | | Fungal sinusitis | 6 | 23.6 | 3.3 | 20.2 | 27.1 | |
| | | Inverted papilloma | 2 | 28.6 | 1.6 | 14.6 | 42.6 | |
| | | Polyps | 12 | 25.1 | 4.4 | 22.3 | 27.9 | |
| | | Cysts | 3 | 32.4 | 3.8 | 23.0 | 41.8 | |
| | | Other lesions | 3 | 21.9 | 1.1 | 19.2 | 24.6 | |
| | | Total | 50 | 25.4 | 4.1 | 24.2 | 26.6 | **0.014** |
| | Maximum sinus area in the frontal plane | Non-specific sinusitis | 24 | 639.2 | 134.5 | 582.5 | 696.0 | **Bonferroni test IP-FS: p = 0.032** |
| | | Fungal sinusitis | 6 | 480.3 | 123.4 | 350.8 | 609.7 | |
| | | Inverted papilloma | 2 | 801.5 | 163.7 | -669.2 | 2272.3 | |
| | | Polyps | 12 | 640.5 | 87.0 | 585.3 | 695.8 | |
| | | Cysts | 3 | 665.3 | 67.1 | 498.6 | 832.0 | |
| | | Other lesions | 3 | 561.3 | 119.9 | 263.5 | 859.0 | |
| | | Total | 50 | 623.9 | 131.9 | 586.4 | 661.3 | **0.023** |

NES = Non-specific sinusitis; FS = fungal sinusitis; OL = other lesions; IP = inverted papilloma. Bonferroni correction has only be applied for the statistically significant values (p <0.05).

and the occupied volume percentage (62.5%) (CC = 0.438, p<0.001). There is no evidence of correlation between the age and the obliteration percentage.

With regard to the sinus dimensions according to the existing pathology, there were no statistically significant differences within the group of healthy sinuses (Table 2). However, in the group of affected sinuses, the medial-lateral width of the sinus in the frontal plane was significantly higher in the cysts group (32.4 mm, CI: 23–41.8 mm) than in the non-specific sinusitis (25.3 mm, CI: 23.8–26.8, p = 0.049), fungal sinusitis (23.6 mm, CI: 20.2–27.1 mm, p = 0.026) and the other lesions groups (21.9 mm, CI: 19.2–24.6 mm, p = 0.019). With regard to the average area of the affected sinus in the frontal plane, the patients with inverted papilloma presented the most elevated area, with 801.5 mm$^2$ (CI: -669-2272.3 mm), significantly higher than for fungal sinusitis, which presented the smallest area out of all of the other pathologies, with 480 mm$^2$ (CI: 350.8–609.7 mm, p = 0.032). There were no differences with regard to the three-dimensional measurements according to the pathology suffered (Fig 4).

The sinus dimensions and the areas, volumes and percentages of occupation were not significantly affected by any other clinical variable, not even by age.

## Discussion

The maxillary sinus is an anatomically complex structure, which has a significant effect on diagnosis and semiology. Technological advances have allowed for the development of systems for the visualisation and three-dimensional representation of this structure. Although the cost of these CBCT-type technologies has decreased, some of the software used for their interpretation can be complex or excessively expensive depending on the levels of precision required

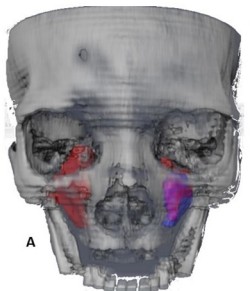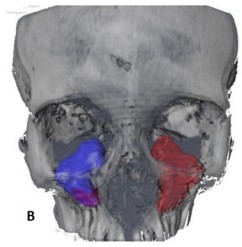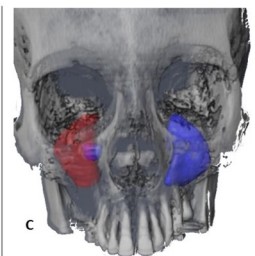

**Fig 4. Volumetric images in frontal view.** Volumetric images in frontal view of the contour and the sinus occupation of 3 representative cases of the sample. In blue, the complete sinus volume, in red the volume which had been pathologically occupied, in purple the volume of residual superposition. A) CBCT16 Total right volume 119.6 mm$^3$, occupied volume 119.6 mm$^3$; total left sinus volume 71.5 mm$^3$, occupied volume 24.6 mm$^3$. B) CBCT10 total right volume 107.4 mm$^3$, occupied volume 16.1 mm$^3$; total left sinus volume 393.8 mm$^3$, occupied volume 393.8 mm$^3$. C) CBCT8 total right volume 241.7 mm$^3$, occupied volume 209.6 mm$^3$; total left volume 111.7 mm$^3$, occupied volume 0 mm$^3$.

[10]. Sinus evaluation through CBCT is useful, especially for otorhinolaryngologists, orthopaedic surgeons and also dentists to evaluate specific alterations like radicular volume loss [11]. The sinus pathology has traditionally been diagnosed through conventional radiology with frontal, Waters and conventional CT projections [12–14]. In this study, CBCTs were used to study and compare healthy and pathological sinuses, to determine the differences between linear measurements, areas and volumes with a specific methodology and to discover the compared efficacy of the volumetric studies with regard to bi-dimensional studies, when calculating the percentage of occupation to measure severity in sinus diseases.

The first studies performed about CT analysed linear measurements and calculated the equivalent volume by means of formulations and algorithms [1]. Ariji et al. 1996 described in their series of cases, an average transversal width of 27 mm (SD = 0.60) and an anteroposterior width of 35.6 (SD = 0.47), which coincides with our measurements of 24.3 mm (SD = 3.8) and 35.5 mm (SD = 3.9), respectively [15].

Very few three-dimensional studies of areas, sinus volumes and obliteration have been carried out. Luz et al. [16] analysed 128 sinuses belonging to 64 patients and detected that the surface area was 39.7 cm$^2$ and the volume average amounted to 17.1 cm$^3$ (14.7 cm$^3$ in our study). 42.2% of all of the sinuses in their study showed a certain level of obliteration compared with 70% in our study. For Sahlstrand et al. [17], the average value of the volume of the maxillary sinus was 15.7 ± 5.3 cm3 and this figure was significantly higher in male than in female subjects. There was no statistically significant correlation between the volume of the maxillary sinus and the age or laterality of patients, as was the case in our study. Gulec et al. found that the average volume of the right maxillary sinus was 13.1 cm$^3$ while the left maxillary sinus was 13.2 cm$^3$, without any significant differences between them [18].

Inflammation continues to be the most prevalent pathology and no relationship with the sinus size or volume has been detected, although significant differences in the width and the maximum area of occupation in the frontal plane have been found [9]. These results are consistent with those of other studies [19,20]. With regard to sinus obliteration, multiple studies suggest that a relationship might exist between the periapical lesions and the irritation of the mucosa of the maxillary sinus [4,21]. Brook et al. [22] demonstrated that between 10 and 12% of all cases of maxillary sinusitis were caused by a dental pathology. In this research, we have excluded the sinuses affected by odontogenic pathology due to the variability and diagnostic complexity, opting to only include sinus pathologies.

As was the case for other authors [18,23], no differences have been found in relation to the volumetric measurements of the sinus and the sex of the patients. However, in other series, smaller [24,25] or larger [17] sizes of sinuses are described for females. With regard to age, the results are very controversial, showing larger sizes in advanced ages [26], with increases in the second and third decades of life [20]. Prabhat et al. [27] were even able to determine the gender by analysing the sinus volume, with a precision of 80.0% in male and 86.7% in female and a global precision rate of 83.3%. Other authors have preconized this forensic technique for gender determination [28]. When compared bilaterally, there are no differences in the sinus size, a fact that has also been proved by previous studies [19,29]. Other studies have confirmed the differences with regard to the sinus volume in relation to the different populations [30,31], however, in our research all of the participants were white. Although it was not an analysed variable in this study, tooth loss does not seem to be a decisive factor in the pneumatization of the maxillary sinus [32].

The automatized evaluation of the sinus volume using certain software programmes allows for fairly reliable volumetric measurements to be obtained, thus avoiding its complex manual determination. Giacomini et al. [33], compared it to the manual segmentation, which was carried out by an experienced radiologist using an automatized standard procedure. This showed average percentage differences between the automated and manual methods of 7.19% ± 5.83% and 6.93% ± 4.29% for the total and sinus volume of the maxillary sinus, respectively. With regard to the inter-observer variations, a high correlation between the measurements obtained by both observers (0.93) has been found in this research, thus coinciding with other authors [34]. In any case, the geometrical or linear method for the dimensional study of the sinus offers a much cheaper, easier and less sophisticated substitute [8]; although if the software is available, the 3D volumetric measurements offer increased precision [35]. The use of linear and bidimensional measurements for the study of sinus occupation and obliteration for the purposes of medical care may be suitable [36]; nonetheless, some authors suggest that the volumetric study slightly improves diagnostic accuracy when compared to clinical scores such as the Sino-Nasal Outcome Test (SNOT-22) and Lund-Mackay and Zinreich modified staging systems [37].

The main limitation of this research is the white origin of all of the patients. Therefore, the conclusions may not be extrapolated to other ethnicities. Another limitation is the inter-observer variability when determining the anatomic limits. In our project, this was at least partly resolved by previously calibrating both operators. The use of different software in different centres could result in variability in the automatized calculations of areas and volumes, therefore all of the radiological studies were analysed using the same system. Among the clinical limitations of this study, the alterations of the anatomic limits in the cases of sinus involvement of malignant tumours must be highlighted, given that this makes it much more difficult for the sinus dimensions, areas and volumes to be determined. That is why malignant lesions were excluded from this research. In terms of implications for future research in this field, other ethnicities must be included to extrapolated in a wider range these specific results. Moreover, if these results are validated, clinical and research protocols must be updated and reformatted.

## Conclusions

The occupation of clinically affected maxillary sinuses is significantly higher than in contralateral sinuses. With regard to the size of the sinus contour depending on the different pathologies, we observed a larger frontal width in the group of patients suffering from cysts and a higher sinus area measured in the frontal plane in the group of patients with inverted

papilloma, without global volumetric differences. In medical terms, the global percentage of occupation determined using the classic manual determination method does not differ from the three-dimensional percentage calculated using specific complex software. Moreover, it must be underlined that the CBCT method, subjects the patient to a lower level of radiation than traditional CT, having the same advantages.

## Author Contributions

**Conceptualization:** Mario Pérez Sayáns, Juan A. Suárez Quintanilla, Pía López Jornet.

**Data curation:** Mario Pérez Sayáns, Cintia M. Chamorro Petronacci, José M. Suárez Peñaranda, Yolanda Guerrero Sánchez.

**Formal analysis:** Mario Pérez Sayáns.

**Investigation:** Mario Pérez Sayáns, Cintia M. Chamorro Petronacci, José M. Suárez Peñaranda, Yolanda Guerrero Sánchez.

**Methodology:** Mario Pérez Sayáns, José M. Suárez Peñaranda, Pía López Jornet, Yolanda Guerrero Sánchez.

**Software:** Francisco Gómez García.

**Supervision:** Juan A. Suárez Quintanilla, Francisco Gómez García.

**Writing – original draft:** Mario Pérez Sayáns.

**Writing – review & editing:** Mario Pérez Sayáns, Juan A. Suárez Quintanilla, José M. Suárez Peñaranda, Pía López Jornet, Francisco Gómez García.

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
