## [Decision Letter · Decision Letter 0]

15 May 2020

PONE-D-20-10893

Volumetric study of the maxillary sinus in patients with sinus pathology

PLOS ONE

Dear Prof Pérez Sayáns,

Thank you for submitting your manuscript to PLOS ONE. After careful consideration, we feel that it has merit but does not fully meet PLOS ONE’s publication criteria as it currently stands. Therefore, we invite you to submit a revised version of the manuscript that addresses the points raised during the review process.

We would appreciate receiving your revised manuscript by Jun 29 2020 11:59PM. To enhance the reproducibility of your results, we recommend that if applicable you deposit your laboratory protocols in protocols.io, where a protocol can be assigned its own identifier (DOI) such that it can be cited independently in the future. For instructions see: http://journals.plos.org/plosone/s/submission-guidelines#loc-laboratory-protocols

We look forward to receiving your revised manuscript.

Kind regards,

Giovanni Cammaroto

Academic Editor

PLOS ONE

Additional Editor Comments (if provided):

Dear Author,

you will see the reviewers have commented on your manuscript and suggested you to revise it.

a revision of your article will be evaluated for eventual publication.

regards

2. Please note that according to our submission guidelines (http://journals.plos.org/plosone/s/submission-guidelines), outmoded terms and potentially stigmatizing labels should be changed to more current, acceptable terminology. For example: “Caucasian” should be changed to “white” or “of [Western] European descent” (as appropriate).

3. In the Methods section please specify the date (month/year) when participants' medical records were reviewed.

Reviewers' comments:

Reviewer's Responses to Questions

**Comments to the Author**

1. Is the manuscript technically sound, and do the data support the conclusions?

Reviewer #1: Yes

Reviewer #2: Yes

2. Has the statistical analysis been performed appropriately and rigorously? 

Reviewer #1: Yes

Reviewer #2: Yes

3. Have the authors made all data underlying the findings in their manuscript fully available?

Reviewer #1: Yes

Reviewer #2: Yes

4. Is the manuscript presented in an intelligible fashion and written in standard English?

Reviewer #1: Yes

Reviewer #2: Yes

5. Review Comments to the Author

Reviewer #1: I thank the editor who gave me the opportunity to review this interesting article about the volumetric study of the maxillary sinus in patients with sinus pathology.

Introduction

(SNOT-22) [5] and Lund-Mackay [6] and Zinreich are clinical scores and not traditional techniques.

Statistical analysis

In which case did you use the chi-square test?

Result

In Table 1, which statistical test did you use to compare the volumes of the sinuses (the ANOVA cannot be used) ???

In table 2, specify that you have written the Bonferroni correction only for the statistically significant values (p <0.05)

Discussion

You have written a lot about the use of CBCT for 3D measurement, I recommend adding these 2 articles that instead talk about the traditional CT system for navigation and 3D reconstruction, so you can do a comparison

Galletti B, Gazia F, Galletti C, Galletti F. Endoscopic treatment of a periorbital fat herniation caused by spontaneous solution of continuity of the papyracea lamina. BMJ Case Rep. 2019 Apr 30;12(4). pii: e229376.

Galletti B, Gazia F, Freni F, Sireci F, Galletti F. Endoscopic sinus surgery with and without computer assisted navigation: A retrospective study. Auris Nasus Larynx. 2019 Aug;46(4):520-525.

Another interesting article to mention about CBCT and the maxillary sinus is the following

Lo Giudice A, Galletti C, Gay-Escoda C, Leonardi R. CBCT assessment of radicular volume loss after rapid maxillary expansion: A systematic review. J Clin Exp Dent. 2018 May 1;10(5):e484-e494.

Basically the article says nothing new compared to that already present in the literature. The authors underline how the CBCT method, with particular 3D reconstruction software, can have economic advantages, compared to conventional techniques. I would give more emphasis to the conclusions, underlining that the CB method subjects the patient to a lower number of radiation than traditional CT, having the same advantages.

Reviewer #2: I found this manuscript apropiate, well designed and clearly exposed, for specialists and for non-specialists. Your results and discussion address the hypothesis and objectives, that are of interest in this field.

I suggest just to review a few details, or solve some doubts I have, to improve the final manuscript, as follows:

- I would appreciate if you clarify if the term "studied element" in the section of exclusion and inclusion and exclusion criteria, refers to the "Unit of Analysis". In this case, it could be a better term.

- Could you specify the voxel size and FOV of the CBCT used?

- It's not clear enough how did you defined an affected sinus by dental pathology reason, that you finally excluded from the sample. I suggest just to clarify this point.

- I suggest to include a comment in discussion about implications for future research in this field.

- I find slightly contradictory the comment in discussion about the "reduced sample size" considering that your initial sample could be greater, and you reduced it by calculating a sample size properly.

- I suggest to write a final conclusion, that is optional, to clearly resume your findings if you find it possible and interesting.

Thank you dear college.

6. PLOS authors have the option to publish the peer review history of their article (what does this mean?). If published, this will include your full peer review and any attached files.

Reviewer #1: No

Reviewer #2: No

---

## [Author Response · Author response to Decision Letter 0]

20 May 2020

May 17, 2020

 Giovanni Cammaroto

Academic Editor

PLOS ONE

Dear Editor

We would like to submit the revised manuscript (PONE-D-20-10893) entitled “Volumetric study of the maxillary sinus in patients with sinus pathology” to the Editorial Board of PLOS ONE. All the changes have been highlighted in bold yellow along the manuscript.

We would like to thank the reviewers for their thoughtful review of the manuscript. We agree with the comments and we have revised our manuscript accordingly. 

After completion of the suggested edits, the revised manuscript has benefitted from an improvement in the overall presentation and clarity. 

Thank you very much for your attention and we look forward to hearing from you at your earliest convenience. 

Sincerely Yours, 

Prof. Dr. Mario Pérez-Sayáns

We have checked carefully all the documents and adapted the manuscript accordingly

2. Please note that according to our submission guidelines (http://journals.plos.org/plosone/s/submission-guidelines), outmoded terms and potentially stigmatizing labels should be changed to more current, acceptable terminology. For example: “Caucasian” should be changed to “white” or “of [Western] European descent” (as appropriate).

Outmoded terms have been changed to more current terminology.

3. In the Methods section please specify the date (month/year) when participants' medical records were reviewed.

Date has been added.

Reviewer #1: I thank the editor who gave me the opportunity to review this interesting article about the volumetric study of the maxillary sinus in patients with sinus pathology.

Introduction

(SNOT-22) [5] and Lund-Mackay [6] and Zinreich are clinical scores and not traditional techniques.

We thank the reviewer this consideration and it has been modified.

Statistical analysis

In which case did you use the chi-square test?

We have completed the explanation as follows: “Contingency tables were drawn up using the Chi-squared test to compare categorical variables”

Result

In Table 1, which statistical test did you use to compare the volumes of the sinuses (the ANOVA cannot be used) ???

Thank you very much for the appreciation. We used the t-student test for independent samples to compare means. Although for wide series we could assume the same p-value. This information has been added to material and methods section

In table 2, specify that you have written the Bonferroni correction only for the statistically significant values (p <0.05).

We appreciate this comment and we have added this information.

Discussion

You have written a lot about the use of CBCT for 3D measurement, I recommend adding these 2 articles that instead talk about the traditional CT system for navigation and 3D reconstruction, so you can do a comparison

Galletti B, Gazia F, Galletti C, Galletti F. Endoscopic treatment of a periorbital fat herniation caused by spontaneous solution of continuity of the papyracea lamina. BMJ Case Rep. 2019 Apr 30;12(4). pii: e229376.

Galletti B, Gazia F, Freni F, Sireci F, Galletti F. Endoscopic sinus surgery with and without computer assisted navigation: A retrospective study. Auris Nasus Larynx. 2019 Aug;46(4):520-525.

Another interesting article to mention about CBCT and the maxillary sinus is the following

Lo Giudice A, Galletti C, Gay-Escoda C, Leonardi R. CBCT assessment of radicular volume loss after rapid maxillary expansion: A systematic review. J Clin Exp Dent. 2018 May 1;10(5):e484-e494.

We thank the reviewer to these interesting papers. They have been added and accordingly cited in the discussion section in order to improve this specific section.

Basically the article says nothing new compared to that already present in the literature. The authors underline how the CBCT method, with particular 3D reconstruction software, can have economic advantages, compared to conventional techniques. I would give more emphasis to the conclusions, underlining that the CB method subjects the patient to a lower number of radiation than traditional CT, having the same advantages.

This is a good appreciation and we have added it to the conclusions 

Reviewer #2: I found this manuscript appropriate, well designed and clearly exposed, for specialists and for non-specialists. Your results and discussion address the hypothesis and objectives, that are of interest in this field.

I suggest just to review a few details, or solve some doubts I have, to improve the final manuscript, as follows:

- I would appreciate if you clarify if the term "studied element" in the section of exclusion and inclusion and exclusion criteria, refers to the "Unit of Analysis". In this case, it could be a better term.

We are very sorry for the misunderstanding, of course the correct form is Unit of analysis and it has been changed accordingly.

- Could you specify the voxel size and FOV of the CBCT used?

Thank you very much for your comment, we have added this information in Material and Methods Section.

Due to the deep anatomical and structural analysis of the radiological studies the values of FOV were high, from 170-229 mm. Regarding the voxel size used, it was between 0.3-0.4 as a general range, adapted to each specific case, so there was absolute precision when calculating the volume.

- It's not clear enough how did you defined an affected sinus by dental pathology reason, that you finally excluded from the sample. I suggest just to clarify this point.

Due to the high and sometimes reversible morbidity, specific characteristics and also the behaviour of odontogenic sinus pathology we have excluded from this study lesions derived from radicular pathology: radicular cysts, periodontal or lateral cysts and other inflammatory lesions developed from a dental destruction/cavities or periodontitis.

- I suggest to include a comment in discussion about implications for future research in this field.

We appreciate this suggestion and we have added a comment addressed to this

- I find slightly contradictory the comment in discussion about the "reduced sample size" considering that your initial sample could be greater, and you reduced it by calculating a sample size properly.

Thank you very much for your comment, in fact it has been poorly expressed. We tried to compare with other bigger samples in terms of sinus pathology studies. We have deleted this comment.

- I suggest to write a final conclusion, that is optional, to clearly resume your findings if you find it possible and interesting.

A conclusion section has been added after the discussion by adding also specific suggestions from other reviewers who have given the same recommendation.

Thank you dear college.

We have uploaded the figures to PACE system and after the checking, they have been submitted with the reviewed version of the manuscript to the PLOS ONE system.

---

## [Decision Letter · Decision Letter 1]

5 Jun 2020

Volumetric study of the maxillary sinus in patients with sinus pathology

PONE-D-20-10893R1

Dear Dr. Pérez Sayáns,

We’re pleased to inform you that your manuscript has been judged scientifically suitable for publication and will be formally accepted for publication once it meets all outstanding technical requirements.

Kind regards,

Giovanni Cammaroto

Academic Editor

PLOS ONE

Reviewers' comments:

Reviewer's Responses to Questions

**Comments to the Author**

1. If the authors have adequately addressed your comments raised in a previous round of review and you feel that this manuscript is now acceptable for publication, you may indicate that here to bypass the “Comments to the Author” section, enter your conflict of interest statement in the “Confidential to Editor” section, and submit your "Accept" recommendation.

Reviewer #1: All comments have been addressed

Reviewer #2: All comments have been addressed

2. Is the manuscript technically sound, and do the data support the conclusions?

Reviewer #1: Yes

Reviewer #2: (No Response)

3. Has the statistical analysis been performed appropriately and rigorously? 

Reviewer #1: Yes

Reviewer #2: (No Response)

4. Have the authors made all data underlying the findings in their manuscript fully available?

Reviewer #1: Yes

Reviewer #2: (No Response)

5. Is the manuscript presented in an intelligible fashion and written in standard English?

Reviewer #1: Yes

Reviewer #2: (No Response)

6. Review Comments to the Author

Reviewer #1: The authors have corrected the paper according to the reviewer's suggestions. So, i think the work can be published after having corrected it as requested.

Reviewer #2: Thank you for adressing the changes I suggested in order to improve the final paper. I find it now properly modified and appropiate to be published.

7. PLOS authors have the option to publish the peer review history of their article (what does this mean?). If published, this will include your full peer review and any attached files.

Reviewer #1: No

Reviewer #2: No

---

## [Editor Report · Acceptance letter]

9 Jun 2020

PONE-D-20-10893R1 

Volumetric study of the maxillary sinus in patients with sinus pathology 

Dear Dr. Pérez Sayáns:

I'm pleased to inform you that your manuscript has been deemed suitable for publication in PLOS ONE. Congratulations! Your manuscript is now with our production department. 

Kind regards, 

on behalf of

Dr. Giovanni Cammaroto 

Academic Editor

PLOS ONE